# Democracy, Capacity, and the Implementation of Laws Protecting Human Rights

**David Cingranelli [1,*], Skip Mark [2] and Almira Sadykova-DuMond [1]**

[1]  Department of Political Science, Binghamton University, Binghamton, NY 13902, USA
[2]  Department of Political Science, University of Rhode Island, Kingston, RI 02881, USA
*   Correspondence: davidc@binghamton.edu

**Abstract:** We analyze the cross-national and cross-temporal variation in the presence or absence of domestic compliance gaps for three different human rights: the right to a fair trial, children's rights, and the right of workers to form unions. Besides constitutional provisions, which have been the focus of previous research on the de jure-de facto compliance gap, statutes, executive actions, and judicial decisions all can contain promises by domestic politicians to protect human rights. Our indicator of whether legal protection exists and how strong it is reflects the many ways states make human rights legal commitments to their citizens. Our findings show that (a) the probability of promise-keeping and the effects of combinations of accountability and capacity are different for each right; (b) strong laws are a necessary but not sufficient condition for effective protection of rights; (c) treaty participation does not affect the probability of promise-keeping for any right; (d) promise-keeping for one right predicted promise-keeping for other rights. For all rights, the number of countries with gaps grew between 1994 and 2008 and then declined between 2008 and 2019. An important inference from our findings is that international treaties may only be effective when ratifiers are willing to change their domestic laws to be consistent with international norms. One counterintuitive policy implication of our findings is that democratizing low-capacity authoritarian states may lead to more violations of some human rights.

**Keywords:** implementation; human rights; fair trial; union; union rights; children's rights; treaties; regime type; accountability; state capacity

## 1. Introduction

Faithful implementation of a domestic law or set of laws protecting internationally recognized human rights requires that legal promises to protect a right be consistent with the efforts the government makes. For most nations and most rights, national promises are equal to or stronger than the actual efforts made. The literature sometimes refers to such incongruencies as de jure/de facto gaps (Voigt 2019). Domestic compliance gaps are common in all policy areas including human rights policies (Blume and Voigt 2007; Law and Versteeg 2013; Mataic and Finke 2019; Finke and Mataic 2018).

Research has shown that more democracy and state capacity reduce compliance gaps (e.g., Anaya-Munoz and Murdie 2021; Cole 2015; Englehart 2009). Democratic states hold regular elections, making them more accountable to what their citizens want. In the following discussion, we use the terms "accountable" and "democratic" interchangeably. More capable states can accomplish the goals they set. In this article, we analyze the cross-national and cross-temporal variation in the presence or absence of domestic compliance gaps for three human rights: the right to a fair trial, children's rights, and the right of workers to form unions. Seeking maximum generalizability, our goal was to test whether a single theoretical model could explain variation in domestic compliance with national promises for all human rights.

Most research on the domestic compliance gap has focused on the correspondence between constitutional promises and national practices. This research expands the conception

and measurement of domestic promises to include many ways other than constitutional promises that domestic politicians pledge to protect human rights. While the inclusion of a constitutional provision in a nation's bill of rights is evidence of a promise, the absence of a constitutional provision is not evidence of the absence of a promise. Domestic promises also can be made through legislative statutes, executive actions, and judicial decisions. For example, the right to a fair trial is protected by the Sixth Amendment to the US constitution, but children's rights and union rights are not constitutionally protected. Both are protected by US laws, reinforced by subsequent court decisions and administrative rulings.

We find some consistencies across models explaining compliance gaps for the right to a fair trial, children's rights, and the right of workers to form unions. As expected, states with both a high level of accountability and a high level of capacity are most likely to keep their domestic rights promises. However, the effects of combinations of accountability and capacity on the probability of promise-keeping were different for each of the three rights examined. For children's rights, democratic states having high accountability, but low capacity were more likely to break their domestic promises than autocratic, low accountability states. Similarly, changes in capacity for unaccountable states did not affect domestic compliance gaps for the right to a fair trial. Therefore, one theoretical model stipulating the effect of accountability and capacity on domestic compliance does not fit all rights.

Tests of our hypotheses produced other important findings. First, ratification of relevant treaties did not affect the probability of a domestic compliance gap for any of the rights. Second, internal political violence increases the probability of a domestic compliance gap for all three rights. Third, if a government kept its domestic promises for one right, it was likely to keep its promises for the other two rights. This last finding supports the common conjecture that human rights are interdependent and mutually reinforcing (Effeh 2007; Quane 2012; Donnelly 2013).

There were some other noteworthy patterns worthy of future exploration. We show that the probability of promise-keeping is different for each right and that strong laws are a necessary but not sufficient condition for the effective protection of human rights. While this pattern is not surprising, when combined with our treaty ratification findings, it does suggest that domestic laws may be more important than treaty participation as a predictor of state protections of human rights. Other patterns present puzzles requiring further research. For example, for the three rights examined, the number of countries with gaps grew between 1994 and 2008, and then declined between 2008 and 2019. Moreover, the probability of a gap was lowest for union rights, corroborating research showing that domestic laws protecting union rights and other group rights tend to be more faithfully implemented than individual rights (Barry et al. 2022; Law and Versteeg 2013).

## 2. Theory and Literature Review

Evaluating domestic compliance requires at least four separate steps. First, one must interpret what the international norm demands of national politicians. Second, one must compare the domestic law protecting a right with the international norm. A strong domestic law promises to fulfill all international requirements, while a weaker law promises some, but not all, of what the international norm requires. Third, one must assess the strength of the domestic practice. Strong practices fully implement international requirements. Weak practices implement some, but not all, international requirements. Finally, one must compare each nation's promises with its practices to assess congruence or non-congruence.

As is common in the study of human rights, the theory of why some states tend to keep their domestic promises rests on a rational choice framework. Within that framework, a principal-agent perspective is most useful for thinking about compliance with domestic laws protecting human rights (Englehart 2009; Cingranelli et al. 2014). Citizens prefer a high level of domestic promises and actual protection of all internationally recognized human rights but must depend upon politicians and bureaucrats to get what they want. The existence of a domestic compliance gap results from strategic decisions by politicians,

who are rational actors deciding whether keeping their domestic promises would help them survive in national political office. Politicians choose to keep their promises for a particular right if they believe that the domestic and international benefits of keeping their promises outweigh the costs of breaking them. Due to inevitable agency loss in the relationship between citizen demands and the decisions of politicians, some degree of domestic compliance gap is likely.

The best way to make politicians more willing to protect human rights is to make them accountable to citizen demands, usually through regular fair elections and universal adult suffrage. A high degree of democracy and strong human rights protection is expected to be closely connected. Politicians also are rational actors with multiple goals. In democracies, they are more willing to protect human rights because institutional constraints make them more accountable to their citizens. At a minimum, citizens can punish politicians who fail to keep their promises by voting against them. Therefore, highly democratic states should be less likely to have any domestic compliance gap. The idea that democratic institutions will promote good human rights practices is well supported by research findings. Institutions that promote accountability are the key to reducing agency loss in the relationship between citizens and politicians.

The decisions by politicians to do what their citizens want may be constrained in some states by the demands of other international actors. For example, some democratic nations are members of powerful regional governments such as the European Union. Human rights laws enacted by the European Union are binding on member governments preventing politicians of member states from being as responsive to citizen demands as they would like. Even participation in powerful intergovernmental organizations, such as the International Monetary Fund (IMF), faces constraints that prevent them from doing what their citizens want. For a more extensive discussion of the "multiple principals, common agent problem", see Abouharb et al. (2019).

Politicians who are accountable to their citizens still must depend upon lower-level members of the executive branch to implement their decisions. They must devote some effort to monitoring the behavior of police, prison guards, soldiers, and other bureaucratic agents. Bureaucrats, akin to politicians, are rational actors with multiple goals. In general, they prefer to devote little effort to protecting human rights. State resources must be expended to limit bureaucratic discretion. Even if the politicians of low-capacity states promise effective protections, they may not be able to execute the policies they enact. They may lack the resources necessary to solve the policy problem and to control the discretion of implementing officials. In their cross-national study of sexual violence, Butler et al. (2007) found that sexual violence was lower in highly capable states because agents of the state were subject to tighter control. It follows that some portion of the domestic compliance gap also results from inevitable agency loss in the principal-agent relationship between politicians and bureaucrats.

The politicians of strong states are more able to do what their citizens want. A state has a high degree of capacity if its politicians can effectively implement state policies. While a strong state could choose to violate specific human rights, the literature shows that more capable states tend to provide stronger protection of many human rights. Some studies have explicitly emphasized the role of state capacity (Anaya-Munoz and Murdie 2021; Cole 2015; Cingranelli et al. 2014; Englehart 2009) as necessary for human rights protection. Posner (2014, p. 148) even argues that providing foreign aid to increase capacity in less economically developed countries, even autocratic countries, is the best way to improve human rights practices. From the principal-agent perspective, state capacity is the key to reducing agency loss in the relationship between politicians as principals and soldiers, police, prison guards, and other bureaucrats as agents.

In her seminal work on the constitutional compliance gap for physical integrity rights, Keith (2001, 2002) found that the presence of an independent judiciary was an important factor in reducing it. This finding has been supported by subsequent research (Melton and Ginsburg 2014; Dietrich and Crabtree 2019; Berggren and Gutmann 2020). Law and

Versteeg (2013) examined the congruence between promises made in national constitutions to protect a wide variety of human rights. Their work identified countries having "sham constitutions". Sham constitutions make promises to protect rights, but implement few, if any, of their promises. They examined three categories of constitutional rights–personal integrity rights, civil and political freedoms, and socioeconomic and group rights for 167 countries, finding that 23.4% of them had sham constitutions. On average, constitutional promises regarding protections of physical integrity and civil liberties had become stronger over time, while actual protections had increased a little for physical integrity rights and had declined for civil liberties. Both promises and practices gradually improved only for socioeconomic and group rights. From the perspective of theory development, one of Law and Versteeg's (2013) most important findings was that rights allowing for the formation of groups such as unions were more likely to be enforced than rights empowering individuals only. They argued that, once groups were allowed to form, taking away their right to group formation was more costly to the government than denying individual rights (see also Barry et al. 2022). Autocratic, unaccountable states and poor, less capable states were overrepresented among those with sham constitutions.

Research on the gap between religious freedoms promised in constitutions and the actual amount of protection states provided (Finke and Mataic 2018; Chilton and Versteeg 2016) has yielded similar results. More accountable and capable states were more likely to adhere to their constitutional promises. Finke and Mataic (2018) also examined the trend in the frequency of gaps between 1990 and 2008. Although the percentage of countries promising religious freedom remained relatively flat, the percentage of countries breaking promises increased. From 1990 to 2008, the more recent the year, the greater the likelihood of gaps between constitutional promises of freedom of religion and actual practices. This finding is consistent with substantial literature indicating that the human rights compliance or implementation gap is a persistent and growing problem (Carraro 2019).

The literature on compliance gaps has also asked whether countries can be classified as either consistent promise keepers or promise breakers. If some can be classified as promise-keepers, then domestic compliance with the laws protecting one right by a country should predict compliance with other rights. Some research suggests that the existence of international compliance gaps for some rights is a strong predictor of international compliance gaps for other rights (Cingranelli and Filippov 2020). In contrast, Blume and Voigt (2007) show that there is inconsistent implementation across different constitutional rights. Of the 136 countries included in their study, 20 were inconsistent in keeping their constitutional promises in all four categories of rights they examined, and another 26 kept their domestic promises in all four. Most countries had a mixed performance.

## 3. Fair Trial Rights, Children's Rights, and Union Rights

We chose to examine the right to a fair trial, children's rights, and union rights partly because of data availability. We needed measures of whether rights were protected by domestic laws and whether those laws are faithfully implemented. There are few indicators available of the strength of domestic human rights promises other than those contained in national constitutions. Furthermore, we wanted to choose a set of rights with potentially different causal processes. Children's rights protect humans who cannot vote and are unusually vulnerable. Union rights are group rights that are typically advocated by organized labor unions. The right to a fair trial is much like a physical integrity right, in this case protecting people accused of criminal offenses from bodily harm potentially inflicted by arbitrary and capricious action by the state. Together these rights help us explore the general applicability of the theoretical arguments.

The right to a fair trial, akin to physical integrity rights, is an absolute right rather than a conditional right. All states are expected to protect this right regardless of their capacity. The requirements for the right to a fair trial and the right to physical integrity are rooted in the International Covenant on Civil and Political Rights (ICCPR). Fair trial is not mentioned in the International Covenant on Economic, Social, and Cultural Rights (ICESCR). Many

national governments model their fair trial laws according to requirements stipulated in the ICCPR. Thus, when compared with the other two rights included in this analysis, the strength of promises to protect the right to a fair trial would be expected to be greater than the promises made to protect the other two rights. As a result, the frequency of domestic compliance gaps would be greater. As will be shown in Table 1 below, both expectations were confirmed.

Though the right to a fair trial is centrally important in the fields of law, sociology, criminology, and political science, there are few comparative cross-national indicators of the protection of this right by states. This is the first large-scale, comparative study to examine the effects of capacity and accountability on the domestic compliance gap for the right to a fair trial. Hathaway (2002) conducted the only large-scale, comparative, cross-national analysis of government practices relative to the international norm that everyone charged with a criminal offense should receive a fair trial. However, her main research question was whether states that had ratified the ICCPR were less likely to have an international compliance gap. She concluded that ratification of the ICCPR was not associated with fairer trials. She did not examine the effects of domestic laws, state capacity, or democratic institutions on national fair trial practices.

The other two rights—children's rights and trade union rights—are usually treated as social and economic rights. However, placing them into any category is problematic. The right of children to be protected from exploitation in employment and the right of workers to unionize were both recognized by the International Labor Organization in Conventions that took effect before the ICCPR and ICESCR were even adopted by the United Nations. For children's rights, expectations for states were recognized in the ICESCR and later elaborated in the UN Convention on the Rights of the Child, the most widely ratified human rights treaty in history.

That both rights are included in the ICESCR is significant because international norms in that convention are not absolute; they are conditional, assigning higher expectations for promises and practices to high-capacity states. The key provision is contained in Article 2, stipulating that all states should protect economic and social rights as much as they can considering their available resources. Due to this conditional language, the international norms for protecting children's rights and trade union rights are relative to state capacity. They allow low-capacity Member States to make modest promises in their domestic laws and modest efforts to protect these rights without risking international rebuke. In contrast, the ILO contends in its "Declaration on Fundamental Principles and Rights at Work" that all states regardless of their capacity should abolish child labor and protect the right of workers to unionize.

Research on government protection of children's rights suggests that high levels of capacity or accountability independently led to stronger protections. Violations of children's rights are most common in the poorest countries, so increasing state capacity is one of the best mechanisms for improving government protection of children's rights (Asadullah and Savoia 2018; Edmonds and Schady 2012). States with more capacity are likely to protect children's rights because, on average, they provide more funding for education and more substantial income transfers to poor families (Alber 2012; Crippin 2020). More democracy also is associated with stronger protections of children's rights (Kabeer et al. 2012). None of the prior comparative studies examined the effects of domestic laws on national practices.

Similarly, research on the rights of workers to form unions shows that high levels of capacity or accountability independently lead to stronger protections (Mosley and Uno 2007; Berliner et al. 2015; Barry et al. 2022; Blanton et al. 2015). Democratic institutions provide labor with the access and voice necessary to exercise political influence (Blanton et al. 2015). More state capacity allows for the progressive realization of all economic rights including worker rights (Mosley and Uno 2007). None of the prior comparative studies examined the effects of domestic laws on national practices or the gap between domestic laws and practices.

### 4. Hypotheses

Are democracy and capacity independent predictors? Most research on the international compliance gap in political science has focused on physical integrity rights and theorized the independent effects of accountability and capacity on congruence. While most studies have emphasized the positive effects of democracy on compliance, some research finds that, because of principal-agent problems in weak states, more capable states provide greater respect for human rights (Butler et al. 2007; Englehart 2009).

Some studies suggest that the relationship between capacity and accountability is conditional. The effect of greater capacity can have both negative and positive effects depending on the accountability level (Powell and Staton 2009; Anaya-Muñoz 2019). Capacity mainly magnifies states' incentives to implement or defect from their domestic promises (Zhou 2012). Berliner et al. (2015), for example, found building capacity alone does not contribute to compliance with domestic labor laws in developing countries. Only states where workers had better representation in the political system significantly benefited from state capacity enhancement.

Strong autocratic states, which lack democratic accountability mechanisms, may choose to keep or break their domestic promises with impunity. Research on the effect of state capacity on the international compliance gap for physical integrity rights is consistent with these conjectures, emphasizing that highly accountable, capable states have smaller gaps, but capable, unaccountable states can present a special danger to human rights (Chae 2021; Powell and Staton 2009; Anaya-Muñoz 2019). Anaya-Munoz and Murdie (2021) agree that neither accountability nor capacity independently affects the size of the international compliance gap for physical integrity rights. However, they go a step further arguing that states only provide high levels of protection of physical integrity rights when *both* accountability and capacity are high.

Hypothesis 1 reflects our expectation that accountability will independently reduce a domestic compliance gap without qualifications. Hypothesis 2 emphasizes that state capacity is a double-edged sword. It can cause human rights practices to be much better in highly accountable states or it can make things even worse in unaccountable states (Chae 2021).

**Hypothesis 1 (H1).** *Greater accountability reduces the probability of a domestic compliance gap regardless of the level of capacity.*

**Hypothesis 2 (H2).** *Capacity amplifies the effect of accountability on a domestic compliance gap in either a positive or negative direction.*

Our third hypothesis explicitly examines a plausible alternative explanation of the domestic compliance gap for human rights. A substantial body of research suggests that if a national government ratifies a treaty, its compliance with international norms will be better (Fariss 2018). Ratifying a treaty might even be treated as another indicator of accountability, since, when a state ratifies a treaty, it is committing to doing what its citizens want—committing to protect the treaty-relevant right. Posner (2014) argues that treaty ratification is not a good indicator of a state's commitment to protection, since almost all states have ratified all of the most significant human rights treaties, varying mainly in terms of how quickly they did so. Most previous studies have concluded that the positive treaty effect if there is one, is conditional and complex (Bell et al. 2019; Conrad and Ritter 2019; Simmons 2010).

Why might human rights treaties have no effect on the domestic compliance gap? States may only adopt treaties that do not require domestic changes, or they may adopt with no intention to translate treaty requirements into domestic law (see Carrubba 2005; Downs et al. 1996). For such states, treaties may simply be "window dressing" as states decouple the policy implications from the ratification process, though ratification may

also increase civil society demands for treaty-relevant rights (Conrad and Ritter 2019; Hafner-Burton and Tsutsui 2005).

Others have argued that human rights treaties only are effective in certain situations such as in newly democratizing states, states with weak domestic courts, states with domestic human rights monitoring institutions, or in monist (as opposed to dualist) states. Monist States, such as those in Scandinavia, incorporate treaties directly in their domestic legislation upon ratification. Dualist States, such as the United States and Canada, do not incorporate treaties directly in their domestic legislation but must pass legislation to implement the provisions of the treaty. (Conrad and Ritter 2019; Verdier and Versteeg 2015; Carver 2010; Moravcsik 2000). Thus. the effects of treaties on the domestic compliance gap may be heterogeneous. More time under a treaty may close the gap in democratic states but have no effect on autocracies. More time under a treaty may have different effects in monist and dualist states, in new and old states, in states with strong and weak domestic courts, and in states with and without domestic human rights monitoring institutions. In some types of states, a long time under a treaty may have a direct effect on reducing or increasing the domestic compliance gap. In most states, treaty ratification is likely a necessary, but not sufficient condition for reducing the gap. Investigating the effects of these distinctions and permutations is beyond the scope of this paper, which focuses on the effects of democracy and state capacity. Instead, we examine whether, on average, time under a treaty reduces the domestic compliance gap. We posit a positive relationship, but, recognizing conflicting findings in the literature and the possibility of a delayed effect of treaty ratification, we consider several non-linear possibilities.

**Hypothesis 3 (H3).** *Time under a treaty reduces the probability of a domestic compliance gap for the right(s) protected by that treaty.*

The idea that human rights are interdependent and mutually reinforcing is widely accepted in the subfield (Donnelly 2013), though it is rarely tested. Our last hypothesis posits that domestic compliance gaps for rights are complementary, meaning that they are interdependent and mutually reinforcing. If compliance gaps are complementary, the absence of a domestic compliance gap for one right should predict the absence of a compliance gap for others. Complementarity implies that politicians would choose a consistent strategy to comply with domestic laws or not. As an example of complementarity, protecting union rights can help reduce the economic exploitation of children, because organized labor has incentives to keep children in school and out of the labor market, thereby increasing the bargaining strength and wages of union members (Crippin 2020). Thus, the two rights are interdependent and respect for one reinforces respect for the other. The theory of institutional path dependence also suggests that countries are either implementing all of their domestic human rights promises or not delivering all of them (Cingranelli and Filippov 2020).

However, recent work on physical integrity rights has found that government violations are not complementary; they are substitutable for one another. Leaders substitute different types of physical integrity rights violations to avoid accountability (DeMeritt and Conrad 2019; Payne and Abouharb 2016; DeMeritt 2016). When leaders choose to improve respect for one right such as arbitrary killing, they often simultaneously choose to decrease protection for another such as disappearances (Payne and Abouharb 2016). If compliance gaps are substitutable, the absence of a domestic compliance gap for one right should not predict the absence of a compliance gap for others.

**Hypothesis 4 (H4).** *The existence of a domestic compliance gap for any human right will predict the existence of domestic compliance gaps for other rights.*

## 5. Methods

The dependent variables are dichotomous indicators for each country-year indicating whether the state complied with or broke its promises to protect each right. The four hypotheses are tested using panel data on a global sample of 140 countries covering 1994–2016. The sample consists of 1559 country years where a government has a score greater than ZERO on legal protections. Country-years assigned a score of ZERO for promises were excluded since governments cannot comply with promises they have not made.

A three-equation multivariate probit model was estimated using maximum likelihood. This statistical model assumes the unobservables follow a multivariate distribution with correlations $p_{1,2}$, $p_{1,3}$, and $p_{2,3}$ to be estimated. Estimation is completed using a Geweke-Hajivassiliou-Keane (GHK) smooth recursive simulator to estimate the model (Clark and Reed 2005; Cappellari and Jenkins 2003; Greene 2002, p. 714). Seventy-five simulated draws were estimated. This model has been used to explore foreign policy substitution (Clark and Reed 2005), and target selection in diversionary conflict (Martinez Machain and Rosenberg 2018).

The model estimates the correlation between disturbances instead of making the usual assumption that the disturbances are zero or constant. By interpreting the estimated correlation of disturbances, the results can help determine whether the compliance decisions are substitutes or complements (Alvarez and Nagler 1998; Clark and Reed 2005). A significant Rho coefficient indicates that missing variables from one equation are significantly related to missing variables in another equation, suggesting a similar data-generating process. A positive Rho between two equations indicates that these rights are mutually reinforcing (Hypothesis 4), while a negative coefficient indicates that rights are substitutable. Non-significant Rho indicates that these equations are unrelated, and a single equation model would be more appropriate.

### 5.1. Dependent Variables

Annual scores for the strength of the laws and practices for all three rights were taken from the CIRIGHTS data project (Mark et al. 2022). For the right to form a trade union and respect for children's rights, a score of ZERO indicated no protection both for the law variable and the practice variable. A score of ONE indicated moderate protection, and a score of TWO indicated full protection. For the right to a fair trial the scores for the strength of the law ranged from ZERO (no law protecting the right) to THREE (a strong law protecting the right). The scores for the strength of the practice also ranged from ZERO to THREE. A score of ONE or TWO indicated moderate protection in law and practice. Scores of THREE indicated strong protection in law and practice.

Information about actual government respect for all three rights was taken from the US Department of State *Country Reports on Human Rights Practices* covering the years between 1994 and 2017. Scores measuring the strength of laws protecting various human rights are not available for most rights. For all three rights, content analysis was conducted to convert the text in the *Country Reports* to numbers (see Cingranelli et al. 2022). For more information on how these rights were scored (see Mark et al. 2022) (or visit cirights.com).

Congruence between the strength of the legal protection and actual practices protecting each right is indicated by a dichotomous variable coded for each right each year. For each right, a score of ONE in a country year indicates that the promise and practice are congruent. A score of ZERO indicates noncongruence—a promise broken. A domestic compliance gap for any right can be created in many ways. If a new domestic policy is enacted, but it does not lead to strengthened implementation efforts, then a compliance gap may be created when none previously existed even if practices do not change.

An alternative measurement strategy for analyzing the gap would be to treat the strength of the law as an independent variable explaining the strength of the practice. Our main model measures the existence or absence of the gap directly as the dependent variable. As a test of robustness, we estimated including the strength of the law as an independent variable and the strength of the practice as the dependent variable. The results were consistent with the findings presented here.

*5.2. Independent Variables*

The main independent variables expected to explain the existence of a gap for each of the three rights are democracy (accountability), state capacity, and time under the controlling human rights treaty. To test Hypotheses 1 and 2, state capacity is measured as government revenue divided by GDP. This is a commonly used measure of bureaucratic capacity (Hendrix 2010; Cingranelli et al. 2014). It correlates highly with other measures of state capacity (Hendrix 2010). Scores were taken from the World Bank's World Development Indicators. Accountability was measured using Boix et al.'s (2013) dichotomous measure of democracy. A dichotomous measure is easier to interpret and discuss in interaction models. As an alternative measure of democracy, the PolityIV combined scale which ranges from −10 (autocratic) to 10 (democratic) is used (Marshall et al. 2015). The findings using the PolityIV indicator are consistent.

To test Hypothesis 3, the indicator used was how long a country has participated in a human rights instrument. For the right to a fair trial, the controlling treaty is the International Covenant on Civil and Political Rights (ICCPR) adopted by the United Nations in 1969. For Children's rights, the controlling international law is contained in the United Nations Convention on the Rights of the Child (CRC) adopted in 1989. For the right of workers to join unions, the most pertinent international law is ILO Convention number 87 adopted in 1948. The number of years a country had been a party to each instrument as well as the square and cube of time under the treaty were included as independent variables in the models. This specification assumes a non-linear relationship for both theoretical and methodological reasons. It also accounts for the possibility that when states improve their laws, a short-term gap may occur until practices catch up and the possibility that leaders may keep promises in the short term, but their commitment may wane in the long term. Finally, the inclusion of a cubic spline can improve estimation in panel data where the dependent variable is binary (Beck et al. 1998).

The Rho correlation coefficient from the multivariate probit model was used to test Hypothesis 4. As discussed above (see model specification), positive relationships would support Hypothesis 4 that leader decisions about domestic compliance with rights are complementary and mutually reinforcing. Negative relationships would indicate that leader decisions about compliance with the rights are substitutes.

As control variables, the model includes the log of a country's population (taken from the World Development Indicators) and civil violence magnitude (taken from Marshall (2012)) as factors that might both affect a domestic compliance gap and affect our other covariates of interest. Almost all studies have found that larger population sizes and various indicators of internal political violence independently increase the probability of having a compliance gap.

## 6. Descriptive Statistics

*6.1. Almost All States Promise to Provide Some Protection for All Three Rights*

An examination of the data before conducting the tests of the four hypotheses revealed some interesting patterns described in this section. Table 1 shows the breakdown of 194 Members of the United Nations in 2016 by the strength of the domestic promises they make to protect each right. Similar patterns prevail for all three rights for other years. As shown in the top row of the table, only a few states did not make any promise to protect rights. Only four did not make a substantial domestic promise to protect the right to a fair trial as that right is defined in international law (Afghanistan, Myanmar, Iran, and Saudi Arabia). Even for the other two rights, only a few states did not make any promise of protection. These few cases of "no promises" were the ones excluded from our analysis. Probably because of the unconditional nature of international expectations, national promises were strongest for the right to a fair trial.

**Table 1.** The strength of domestic laws protecting three rights in 2016, N = 194.

| Strength of Domestic Promise | Fair Trial Rights | Union Rights | Children's Rights |
|:---:|:---:|:---:|:---:|
| None | 4 | 18 | 14 |
| Moderate | 59 | 97 | 148 |
| Strong | 131 | 79 | 31 |

### 6.2. Domestic Laws Protecting Human Rights Are a Necessary, but Not Sufficient Condition for Human Rights Protection

Previous research has shown that states rarely protect human rights more strongly than their laws require (Barry et al. 2022). Table 2 shows this was true for the three rights examined here. Only a small percentage of states provided more protection of a human right than the law required—2.7% for the right to form trade unions, 5.5% for the protection of children's rights, and 0.5% for the right to a fair trial. Promise breaking (Laws > Practices) was much more common for protecting children (50.1%) and protecting the right to a fair trial (60.1%) than it was for the right to unionize (32.7%).

**Table 2.** Laws relative to practices, 1981–2016.

|  | Laws > Practices | Practices = Laws | Laws < Practices | Total |
|:---|:---:|:---:|:---:|:---:|
| Right to unionize | 1564 (32.7%) | 3096 (64.7%) | 127 (2.7%) | 4787 |
| Children's rights | 2426 (50.7%) | 2102 (43.9%) | 264 (5.5%) | 4792 |
| Right to a fair trial | 2837 (60.1%) | 1853 (39.3%) | 25 (0.5%) | 4715 |

### 6.3. The Frequency of Domestic Compliance Gaps Is Declining

Figure 1 shows the trends in the frequency of domestic compliance gaps for the three rights examined in this article. As noted above, Mataic and Finke (2019) found that the average size of the domestic compliance gap for freedom of religion had grown between 1990 and 2008. For the three rights examined here, the number of countries with gaps grew between 1994 and 2008 and then declined between 2008 and 2017. Kucera and Sari (2019) in their study on the domestic legal protection of trade union rights in 185 ILO member states also found a declining frequency of gaps between domestic laws protecting collective labor rights and practices in recent times.

As was previously shown in Table 2, the likelihood of a domestic compliance gap was greatest for the right to a fair trial and the smallest for the right to form trade unions throughout the entire 1981–2016 period of this study. One reason for the greater likelihood of a compliance gap for the right to a fair trial than for other rights for all years may be that, as shown in Table 1 above, domestic fair trial promises, on average, were stronger. Keeping strong promises is harder than keeping moderate ones. However, other factors must play an important role. As was shown in Table 1, more states promised full protection of trade union rights than children's rights, but gaps were more likely for children's rights anyway.

Results showing the smallest domestic compliance gaps for trade union rights are consistent with the argument that national governments are most likely to keep their promises for protecting collective rather than individual rights (Chilton and Versteeg 2016). For the right to a fair trial, temporal coverage goes back to 1981. The long-term trend for the right to a fair trial is much the same as the figure above. The frequency of gaps between legal protection and actual government protection increased from 1981 to about 2010 and then declined.

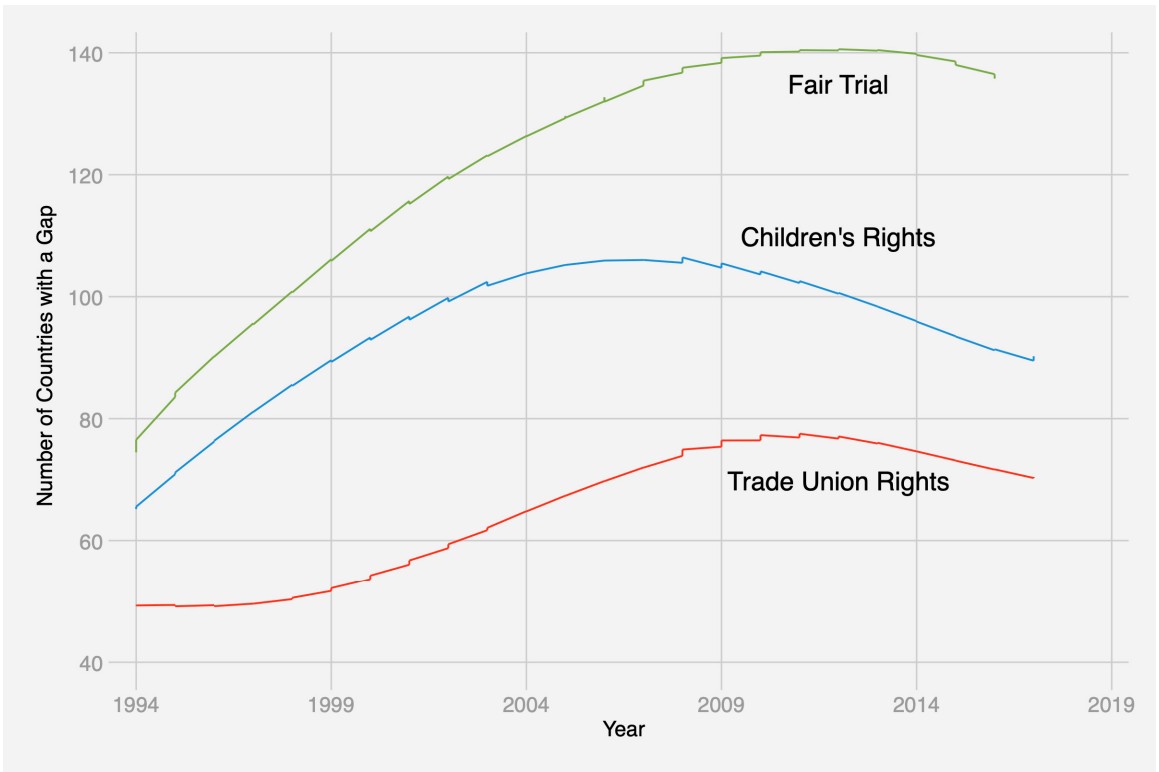

**Figure 1.** Lowess Trends in Domestic Human Rights Gaps over Time.

### 7. Results

Table 3 below shows the results of a naive model (model 1) and our theoretical model (model 2). To keep the results comparable across the three rights, the same model was estimated for each right. The first thing to note is that if accountability and capacity are treated as independent of one another as they are in model 1, they both have a statistically significant negative effect on the domestic compliance gap for all three rights. Higher accountability and state capacity both independently reduce the probability of compliance gaps for all three rights. These findings are consistent with most previous findings in the literature.

In both models, there is no support for Hypothesis 3. Time under an international treaty does not have any effect on the domestic compliance gap. There is also no support for the argument that population size significantly impacts domestic compliance gaps, though previous research has found that it affects the size of the international compliance gap for most rights including physical integrity rights. As expected, civil conflict significantly increases the domestic compliance gap across all three rights.

When examining model 2, it becomes clear that accountability and capacity are not independent of one another and that their conditional relationship is more complex than the literature has recognized to date. The interaction term in all three models is statistically significant, indicating that these variables are not independent of one another but the effect of each is conditional on the values of the other. These coefficients cannot be interpreted by themselves as they are conditional on all constituent elements. Therefore, to correctly interpret the results from model 2 concerning the effects of accountability and capacity as specified in Hypotheses 1 and 2, out-of-sample predictions were generated. The interpretation of interaction terms is often difficult without using graphs (Brambor et al. 2006).

**Table 3.** Multivariate Probit Model.

| | Model 1 | | | Model 2 | | |
|---|---|---|---|---|---|---|
| | **Union** | **Children** | **Fair Trial** | **Union** | **Children** | **Fair Trial** |
| | Eq 1 | Eq 2 | Eq 3 | Eq 1 | Eq 2 | Eq 3 |
| Rho | Union/Child 0.335 *** (0.079) | Child/Trial 0.419 *** (0.082) | Trial/Union 0.595 *** (0.087) | Union/Child 0.305 *** (0.083) | Child/Trial 0.400 *** (0.083) | Trial/Union 0.588 *** (0.087) |
| Capacity × Accountability | | | | −5.717 *** (1.266) | −3.687 *** (1.375) | −2.556 * (1.352) |
| Capacity | −1.550 * (0.908) | −4.302 *** (0.825) | −2.822 *** (0.837) | 2.350 *** (0.848) | −1.793 * (1.017) | −0.985 (0.936) |
| Accountability (0/1) | −0.384 ** (0.185) | −0.381 ** (0.189) | −0.741 *** (0.195) | 0.995 *** (0.352) | 0.529 (0.406) | −0.115 (0.420) |
| Treaty Years | 0.035 (0.034) | 0.010 (0.035) | 0.057 (0.047) | 0.037 (0.034) | 0.007 (0.035) | 0.057 (0.046) |
| Treaty Years$^2$ | −0.002 (0.002) | 0.001 (0.003) | −0.004 (0.003) | −0.002 (0.002) | 0.001 (0.003) | −0.004 (0.003) |
| Treaty Years$^3$ | 0.000 (0.000) | −0.000 (0.000) | 0.000 (0.000) | 0.000 (0.000) | −0.000 (0.000) | 0.000 (0.000) |
| Population (log) | −0.006 (0.054) | −0.014 (0.064) | 0.073 (0.067) | −0.020 (0.052) | −0.024 (0.064) | 0.066 (0.067) |
| Civil Violence | 0.213 ** (0.105) | 0.293 *** (0.112) | 0.366 ** (0.155) | 0.185 * (0.101) | 0.271 ** (0.115) | 0.354 ** (0.157) |
| Constant | 0.372 (0.947) | 1.692 (1.086) | 0.849 (1.185) | −0.263 (0.888) | 1.277 (1.100) | 0.542 (1.180) |
| Observations | 1470 | 1470 | 1470 | 1470 | 1470 | 1470 |
| Wald ~ X$^2$ | | 102.86 *** | | | 111.12 *** | |
| LR ~ X$^2$ Rho$_{12}$ = Rho$_{23}$ = Rho$_{31}$ = 0 | | 215.847 *** | | | 202.804 *** | |

Note: Robust standard errors in parentheses *** $p < 0.01$, ** $p < 0.05$, * $p < 0.1$. Accountability is a 0/1 measure of democracy. State capacity is government revenue divided by GDP.

Figures 2–4 below show out-of-sample predictions for model 2. To produce the graphs, a dataset was generated where all the control variables were held at their mean or mode and the model 2 predictions were used to calculate the probability of a domestic compliance gap (*y*-axis) across all values of state capacity (the *x*-axis) for accountable (democratic) and unaccountable (autocratic) states. The graphs include 95% confidence intervals around those predictions. To evaluate the substantive effects of these comparisons, we pull out the predictions using one standard deviation below (17%) and one standard deviation above (35%) the sample mean of state capacity (25%). Low-capacity states are those where government revenue makes up 17% of GDP while high-capacity states are those with a government revenue of 35% of GDP.

Figures 2–4 show that different combinations of accountability and capacity produced different probabilities of gaps for each of the three rights. Unaccountable, autocratic states such as China and Russia were less likely to break their domestic promises to protect children's rights, were more likely to break their domestic promises to protect union rights and were neither more nor less likely to break their domestic promises to protect the right to a fair trial. This suggests that the role of state capacity in autocratic states varies by the type of rights, sometimes leading to more promise-keeping and at other times leading to more broken promises. For all models, high capacity, accountable states had the highest likelihood of keeping promises, consistent with findings in the literature.

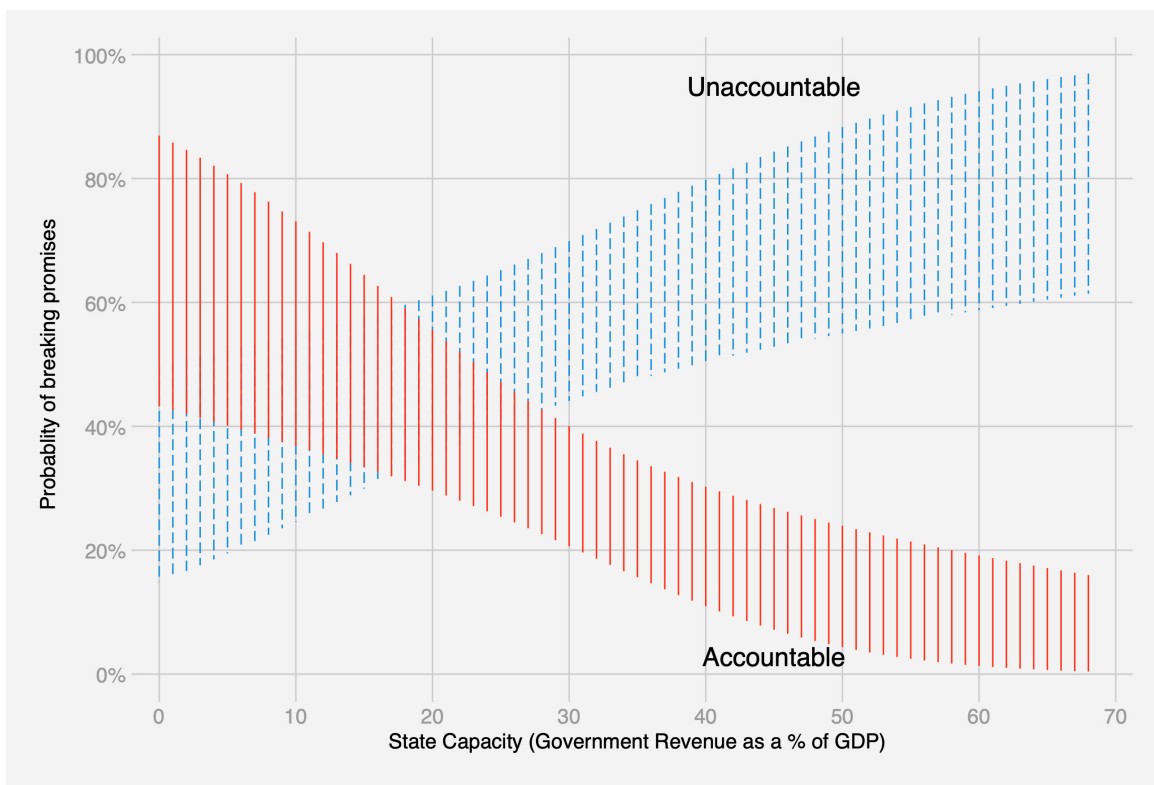

**Figure 2.** Accountability, Capacity, and the Union Gap.

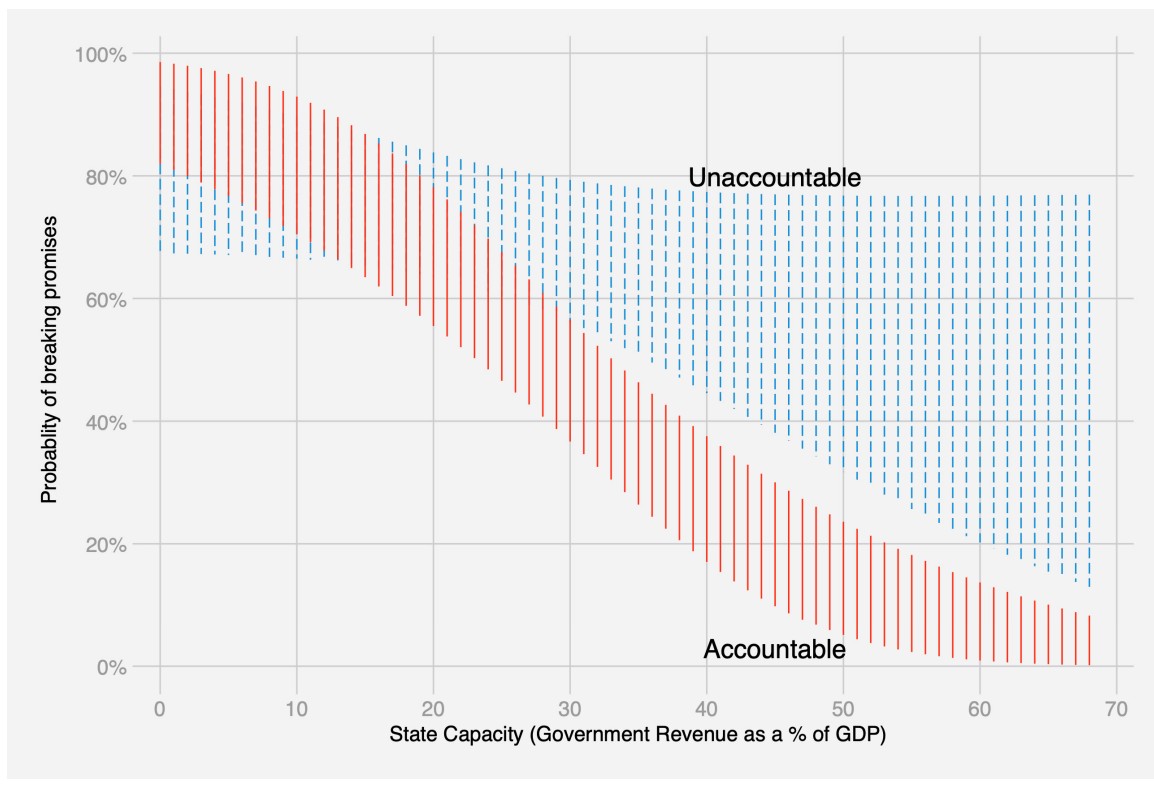

**Figure 3.** Accountability, Capacity, and the Child Rights Gap.

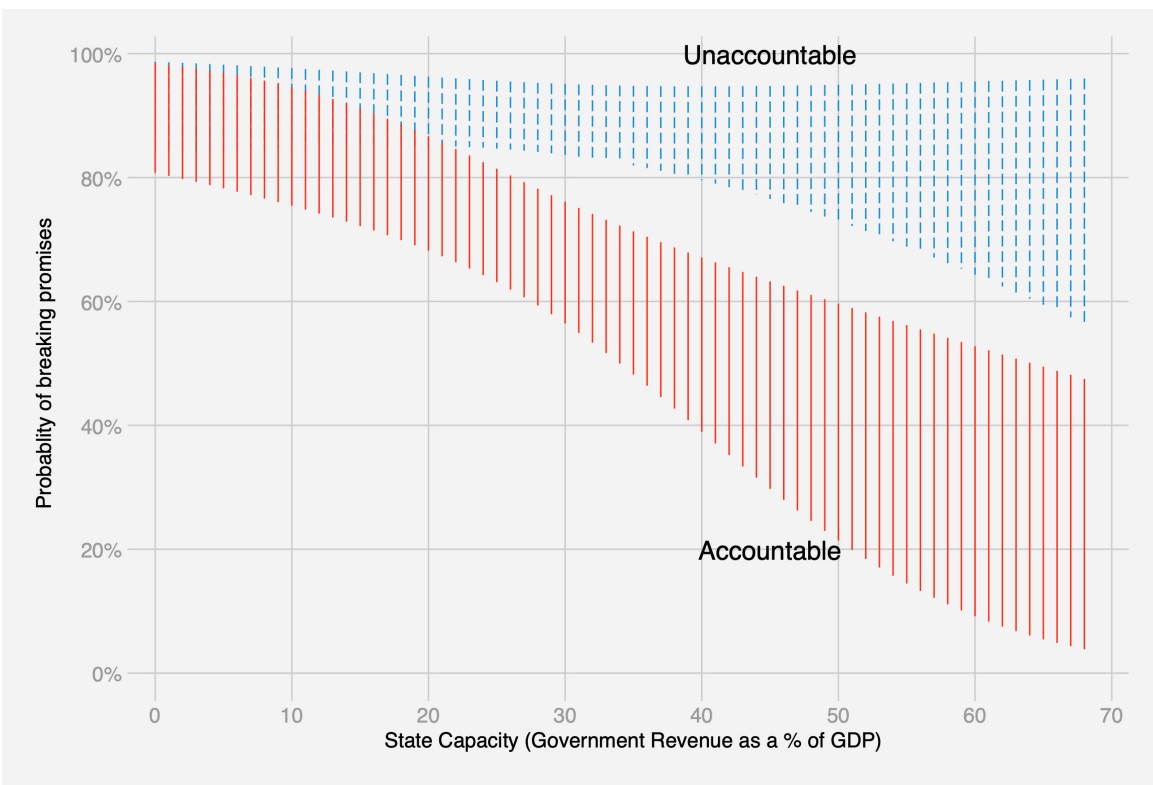

**Figure 4.** Accountability, Capacity, and the Fair Trial Gap.

Low-capacity but accountable states broke some of their domestic human rights promises more often than low-capacity unaccountable states did. Accountable but low-capacity states such as Haiti were more likely to break their domestic promises to protect trade union rights, less likely to break their domestic promises to protect the right to a fair trial, and neither more nor less likely to break their domestic promises to protect children's rights. These findings suggest that efforts to increase accountability in low-capacity states may undermine the protection of some human rights while increasing the protection of others.

Figure 2 graphically presents the predictions from model 2, equation 1, the right to form trade unions. Accountability decreases promise-breaking at an accelerating rate as state capacity increases. For the right to trade unions, high-capacity accountable states have a 24% probability of experiencing a domestic compliance gap. High-capacity unaccountable states on the other hand have a 62% probability of a compliance gap. Low-capacity accountable states are slightly more likely to have a domestic compliance gap (46%) compared to low-capacity unaccountable states (45%) though this difference is not statistically significant. This gap is statistically significant at very low levels of state capacity (10%) where low-capacity accountable states have a 56% probability of a domestic compliance gap compared to 39% for low-capacity unaccountable states.

Figure 3 shows that Increasing state capacity increases the probability of a state keeping its promises to protect children's rights whether the state is democratic (accountable) or authoritarian. The marginal effect of accountability on promise-breaking is zero for low-capacity states, but negative when government revenue exceeds 32% of GDP. High-capacity accountable states are still the least likely to have a compliance gap with a 35% probability. However, increased capacity reduces the likelihood of a compliance gap for children's rights among unaccountable states as well. High-capacity unaccountable states are more likely to keep their promises (65%) regarding children's rights than low-capacity unaccountable states (74%). Low-capacity accountable states (73%) and low-capacity unaccountable states (76%) have similar probabilities of a domestic compliance gap. The gap between democratic

(accountable) and authoritarian (unaccountable) states only emerges at higher levels of state capacity (when government revenue reaches 32% of GDP). At moderate or low levels of state capacity, government accountability does not appear to affect the likelihood of faithful implementation of the law.

Figure 4 shows that capacity augments the effect of democracy so that high-capacity, democratic states are less likely to break their fair trial promises than are low-capacity states regardless of accountability. However, state capacity does not affect the likelihood of having a compliance gap among unaccountable states. Low-capacity unaccountable states have a 93% probability of having a gap compared with a 90% probability for high-capacity unaccountable states. There is no significant difference between low-capacity accountable (82%) and low-capacity unaccountable (93%) states at low levels of state capacity. However, the probability of having a compliance gap is much higher for the right to a fair trial for all combinations of accountability and capacity compared to the other two rights. This suggests that most states do not comply with their domestic laws protecting the right to a fair trial regardless of regime type or state capacity.

The three equations in model 2 allow us to reject hypothesis one. Accountability does not reduce the domestic compliance gap at low levels of state capacity, and for one equation greater accountability increases the domestic compliance gap at low levels of state capacity. They support Hypothesis 2 showing that the interplay between accountability and capacity and their joint effect on states' compliance with their respective domestic human rights laws are far more complex than suggested in Hypothesis 1 or the literature.

Model 2 findings are largely consistent with Hypothesis 2. High capacity augments the effect of accountability, meaning high-capacity, accountable states are less likely to break agreements than are low-capacity states regardless of accountability or human right. The findings from model 2 equation 1 suggest an area of future research that has been largely ignored. In low-capacity states, greater accountability leads to more broken promises for union rights. This suggests that efforts to increase accountability in low-capacity states increase the probability of broken promises for some human rights. For our other two rights, accountability does not affect whether leaders keep domestic promises at low levels of state capacity. The positive effect of accountability on human rights promise-keeping exists only for capable states.

Since interaction terms are symmetric, we can make similar conclusions about the effect of accountability across different levels of capacity as well (Berry et al. 2012). For accountable states, an increase in capacity is always associated with an improvement in promise-keeping. In contrast, we find that the effect of state capacity for unaccountable states can lead to more promise-keeping for some rights and less promise-keeping for others. While it is clear that high-capacity accountable states are the most likely to keep their domestic human rights promises, there seem to be tradeoffs in promise-keeping when unaccountable states improve their state capacity.

Hypothesis 4 is supported by both model 1 and model 2. Rho is statistically significant and positive in both models. This tells us that a gap in one right is positively correlated with a gap in another right. This is consistent with Hypothesis 3 that these rights are mutually reinforcing rather than substitutes. The models suggest that these rights are interdependent and that changes in one right will affect changes in others in the same manner. The strongest correlation occurs between the right to a fair trial and trade union rights. These findings suggest that not all rights are substitutable, and that further work is needed to uncover which rights serve as substitutes for one another and which rights are mutually reinforcing.

Many of the findings reinforce those yielded by some previous studies and, as a result, contribute to the accumulation of knowledge in the human rights subfield. Internal political violence increased the probability of a domestic compliance gap for all three rights. Strong laws are a necessary but not sufficient condition for the strong protection of human rights. All control variables that have been used in previous cross-national studies performed as

was expected. A series of alternative model specifications were examined, and, overall, the findings were consistent with those presented here.

## 8. Conclusions

This research project builds upon a well-established research program focusing on the gap between promised protections of rights contained in national constitutions and actual efforts to protect rights by governments. There is a need for another, related research program focusing on a broader conception of domestic human rights promises that better reflects the many ways states make human rights commitments to their citizens. The results presented here also take us a step closer to a comprehensive theory of why governments keep or break their promises to protect human rights and why promises to protect some rights are more likely to be kept.

Our use of a rational choice, principal-agent perspective is common in the subfield as is the theoretical framework focusing on the interplay between accountability and capacity. Most studies have shown that both a state's level of accountability and its capacity independently reduce the likelihood of a gap between human rights promises and practices. Previous research also suggests that accountability is a more consistent predictor of good performance than capacity. Following that line of argument, we hypothesized that, for all rights, greater accountability would reduce the probability of a domestic compliance gap regardless of the level of capacity (H1). We also hypothesized that capacity would amplify the effect of accountability on a domestic compliance gap either in a positive or negative direction (H2).

The findings show that the theoretical and empirical relationships between accountability, capacity, and the existence of a domestic compliance gap are more complicated than the literature or our hypotheses suggested. They are conditional and interactive as expected. However, the precise nature of the conditionality and interactions of their effects on the probability of having a domestic compliance gap varied depending on the right.

As expected, high-capacity, democratic states were the most likely to keep their domestic human rights promises for all rights. Increasing the capacity of authoritarian states can lead to better human rights promise-keeping for some rights while simultaneously leading to more broken promises for others. Low-capacity, democratic states were more likely to have a gap between their laws and practices protecting union rights than low-capacity, authoritarian states.

One counterintuitive policy implication of our findings is that democratizing low-capacity authoritarian states often leads to *more* violations of some human rights as was the case for union rights. The difficulty of increasing both accountability and capacity simultaneously often leads to a focus on democratization when more could be gained by increasing capacity. Scholars should expand the list of rights we explore and examine how different rights are affected by the accountability-capacity tradeoff. The findings also suggest that focusing on how accountability and capacity improve a limited set of rights might be hiding increased violations of other rights.

We found no evidence that leaders substituted repressive tactics to avoid accountability, a finding that has emerged in recent research on physical integrity rights (Kitagawa and Bell 2022; DeMeritt and Conrad 2019; Payne and Abouharb 2016). This research project extended the previous examinations of complementarity-substitutability to rights other than physical integrity rights. One reason physical integrity violations can be substituted for one another is that they all are tools that governments use to confine or otherwise harm the physical bodies of human beings. In contrast, the leader's logic for protecting fair trial, union, and children's rights are likely to be different in each case. Therefore, the substitution of one type of violation for another would be less useful.

Other patterns in the data present puzzles warranting further investigation. For example, the probability of a gap was lowest for union rights (Figure 1 and Table 2). This finding reinforces the results of research showing that domestic laws protecting union rights and other group rights tend to be more faithfully implemented than individual

rights (Barry et al. 2022; Law and Versteeg 2013). Research on other group rights such as indigenous peoples' rights is necessary to determine whether the relationships observed can be generalized. We also found that domestic laws protecting human rights were a necessary, but not sufficient condition for human rights protection (Table 2), suggesting that more attention to domestic laws protecting human rights is justified. Future research also should examine whether and under what circumstances membership in powerful regional governments and intergovernmental organizations increases the probability of domestic human rights compliance gaps.

Using updated and extended data led to our most intriguing discovery: for the three rights examined, the number of countries with gaps grew between 1994 and 2008, and then declined between 2008 and 2016 (Figure 1). Finke and Mataic (2018) found that between 1990 and 2008 the gap between constitutional promises of freedom of religion and actual practices increased. However, the decline in the frequency of gaps after 2008 was unexpected and requires further investigation. If subsequent research confirms that the frequency of domestic compliance gaps for these and other rights has declined in recent years, the question is "why?".

Perhaps treaties have had more effect on compliance with domestic laws over time. Our findings showed that, on average, time under a treaty did not affect the probability of a domestic compliance gap. They showed that human rights treaties are only effective when ratifiers are willing to change their domestic laws to be more consistent with international norms. Future research should examine the potentially different effects of treaty ratification in different types of states within the framework of democracy and capacity we have emphasized. For example, the effects may be different for democratic and authoritarian monist states. Under some circumstances, more time after ratifying a treaty may lead to stronger national promises, which do matter and matter more as time passes.

**Author Contributions:** Conceptualization, D.C. and S.M.; methodology, S.M.; investigation, A.S.-D.; data curation, D.C., S.M. and A.S.-D.; writing—original draft, D.C. and S.M.; writing—review and editing, D.C., S.M. and A.S.-D.; project administration, D.C. All authors have read and agreed to the published version of the manuscript.

**Funding:** The authors received no external funding for this research.

**Institutional Review Board Statement:** Not applicable.

**Informed Consent Statement:** Not applicable.

**Conflicts of Interest:** The authors declare no conflict of interest.

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
