# Peer review of "Democracy, Capacity, and the Implementation of Laws Protecting Human Rights"

_laws_

Round 1
Reviewer 1 Report
This article is focused on the gap between promised protections of rights contained in national constitutions and actual efforts to protect rights by governments. Its finding “suggests human rights treaties may only be effective when ratifiers are willing to change their domestic laws to be more consistent with international norms”.
In this perspective, I would suggest that the author also considers the systems provided in some regional contexts (e.g., in Europe with the European Union and the European Convention on Human Rights) where the process of ratification and implementation of supranational rights (so the gap between promised protections of rights and the actual efforts to protect them) may be conditioned by the supranational judicial system and the jurisprudence of supra-state Courts (e.g., the Court of Justice of the European Union and the European Court of Human Rights).
Author Response
Reviewer 1 correctly notes that our argument and empirical analysis ignore the role of regional governments (e.g., in Europe with the European Union and the European Convention on Human Rights) where the process of ratification and implementation of supranational rights (so the gap between promised protections of rights and the actual efforts to protect them) may be conditioned by the supranational judicial system and the jurisprudence of supra-state Courts (e.g., the Court of Justice of the European Union and the European Court of Human Rights). This is a point that we, the authors had thought about and discussed while writing our article. However, based on Reviewer 1’s comments, we have acknowledged the issue. See manuscript lines
115-123, 684-686, 704-705
We have also made other small revisions to the Theory and Literature Review Section to clarify our argument.
Reviewer 2 Report
The article is well written and presents contributions to an important discussion. The results are presented clearly but graph design can be improved. Arguments are well engaged and supported by secondary and referenced literature. Overall, the article merits publication as is.
Author Response
We thank Reviewer 2 for his kind comments about our article. R2's only suggestion was that graph design should be improved. All of the graphs in the revised manuscript are now in color, making them easier to read, and Figures 2 and 3 have been modified in other ways to make them more understandable.